# Field-level inference in cosmology

**Florent Leclercq**

CNRS & Sorbonne Université, UMR 7095, Institut d'Astrophysique de Paris,
98 bis boulevard Arago, F-75014 Paris, France

florent.leclercq@iap.fr

## Abstract

These lecture notes delve into field-level inference, a framework offering a robust way to extract more information and avoid biases compared to traditional methods for cosmological data analysis. The core idea is to analyse uncompressed maps to infer underlying physical fields and cosmological parameters. We introduce Bayesian hierarchical field-level models and discuss sampling techniques for exploring complex, high-dimensional posterior distributions. We review the framework that underpins field-level inference. Finally, we highlight some state-of-the-art applications across various cosmological probes, and the growing role of machine learning in enhancing field-level inference capabilities.

# 1  Introduction

## 1.1  What is forward modelling?

Cosmology faces significant challenges arising from the increasing volume and complexity of observational data, and from the concurrent growth of simulations, which are becoming the primary means of expressing theoretical models. Field-level inference is specifically designed to address these challenges.

Making the connection between theory and data requires, on the one hand, predicting some relevant function of the data. On the other hand, it requires measuring the same function by summarising the raw data and usually "correcting" them for certain effects. Forward modelling describes how cosmological parameters and underlying physical fields map to a measurable function of the raw data; for example, cosmological parameters → initial density field → final density field → mock galaxy catalogues → correlation functions. Each step is usually expensive and potentially complicated to describe. At the other end, "backward modelling" assumptions and corrections map the raw data to the same measurable function: correlation functions ← observed galaxy catalogues ← images or spectra ← photons incident on instruments. Each step here is lossy and potentially bias-inducing.

Ideally, forward modelling should be used all the way to predicting photon counts incident on the instruments or, if that is impossible, images or spectra. This is, of course, far from realistic in terms of modelling accuracy and computational cost. Current methodological developments focus on making the connection between theory and data, either by using *any statistical summary* of mock and observed catalogues or by using the *full mock and observed catalogues* themselves to make this connection, without compressing them. The former is known as implicit likelihood inference; the latter is known as field-level inference, which is the subject of these lecture notes. Forward modelling is a crucial aspect of field-level inference.

## 1.2    What is field-level inference?

Field-level inference uses statistical techniques to infer physical fields and cosmological parameters from observed data. It can be broadly defined as any method in which the likelihood is specified *at the level of a full, discretised field* (e.g. a pixelised map), without compression of the corresponding map-level data of any kind. It reconstructs *cosmological maps* as a natural by-product. For these reasons, the likelihood in field-level inference is (almost always) *explicit* and defined at the level of the predicted field. Furthermore, field-level inference problems are very high-dimensional, implying that sampling their posterior distributions requires advanced statistical techniques.

We note that any method that involves compression of the map-level data is not included in this definition. Approaches that first reduce the map to lower-dimensional summaries (whether via usual summary statistics such as correlation functions, statistical compression such as score compression, or neural "optimal" compression methods) are therefore excluded: although they may involve a generative model for the field, the inference itself is ultimately performed in the compressed-data space, not at the level of the field.[1]

Field-level inference provides an alternative to traditional summary-statistics methods, aiming to avoid biases by accounting for complexities that summary statistics may overlook and to extract more information [e.g. 1].

## 1.3    Bayesian signal-processing framework

Field-level inference operates within a Bayesian signal-processing framework, utilising Bayes' theorem for inference. In field-level inference, we are usually not able to compute (or not interested in computing) the Bayesian evidence of the model. Therefore, it is sufficient to work with unnormalised probabilities, and we can write Bayes' theorem as

$$\ln p(\boldsymbol{\theta}|\boldsymbol{d}) = \ln p(\boldsymbol{\theta}) + \ln p(\boldsymbol{d}|\boldsymbol{\theta}) + \text{const}, \tag{1}$$

where $\boldsymbol{d}$ is the observed field, and the vector of target parameters $\boldsymbol{\theta}$ contains a signal field $\boldsymbol{s}$ and possibly a lower-dimensional set of other parameters. Forward modelling thus involves the construction of the log-prior $\ln p(\boldsymbol{\theta})$, the log-likelihood $\ln p(\boldsymbol{d}|\boldsymbol{\theta})$, and the log-posterior $\ln p(\boldsymbol{\theta}|\boldsymbol{d})$ of the model.

## 1.4    Introduction to Bayesian hierarchical models

Bayesian hierarchical models are crucial for tackling complex problems that involve multiple sources of randomness, variability, or uncertainty. The key idea is to split the inference problem into steps, where the full model is composed of a series of sub-models. The Bayesian hierarchical model links the sub-models together, correctly propagating uncertainties in each sub-model from one level to the next.

Bayesian hierarchical modelling is a principled way to include, for instance, systematic errors and selection effects. It also clarifies what must be known or assumed, usually a set of conditional probability distributions. All intermediate sub-models give rise to "latent variables," which are usually not of direct interest. Therefore, Bayesian hierarchical models may

---

[1]Allowing arbitrary compression would render almost every cosmological analysis a "field-level inference", since familiar summaries such as the power spectrum originate from a field representation at some point. In fact, methods achieving maximal compression (reducing the data to as many numbers as there are target parameters) can be regarded as lying at the opposite extreme of what we mean by field-level inference. This distinction is purely operational and does not diminish the utility of compressed-data approaches; it clarifies the scope of the term as used in these lectures.

101  have a very large number of parameters. In field-level inference, Bayesian hierarchical mod-
102  els enable the joint inference of both underlying fields and the cosmological parameters that
103  define their behaviour, offering a robust and self-consistent inference framework.

104      These lecture notes are organised as follows. Section 2 reviews Wiener filtering for Gaus-
105  sian linear models and establishes it as the analytic benchmark against which non-linear ap-
106  proaches are compared. Section 3 reviews the Markov Chain Monte Carlo techniques needed
107  for field-level inference with non-linear models and develops quantitative diagnostics for Markov
108  chains. Section 4 presents the solution to field-level inference with non-linear models. Section
109  5 discusses considerations that arise when sampling a set of additional parameters jointly with
110  the field. In Sections 2, 4 and 5, we exemplify field-level inference in practice, first using a
111  linear model, which is then generalised to a non-linear field-level model with local-type pri-
112  mordial non-Gaussianity. Section 6 reviews some state-of-the-art cosmological applications of
113  field-level inference and outlines directions of current research.

## 2  Bayesian denoising via Wiener filtering for linear models

115  This section introduces initial approaches to field-level inference, focusing on signal recon-
116  struction (Bayesian denoising) for linear models.

### 2.1  Gaussian random fields

118  Gaussian random fields (GRFs) are fundamental ingredients in many field-level models. A GRF
119  is defined as a random vector $x$ with probability density function (pdf) $p(x|\mu, \mathbf{C})$ satisfying

$$-2\ln p(x|\mu, \mathbf{C}) = (x - \mu)^{\mathsf{T}}\mathbf{C}^{-1}(x - \mu) + \ln|2\pi\mathbf{C}|, \tag{2}$$

120  for any vector $\mu$ (the mean) and any real,[2] symmetric, positive-definite matrix $\mathbf{C}$ (the covari-
121  ance matrix).[3]

122      **Integration by differentiation.** A convenient trick for evaluating Gaussian integrals can
123  be referred to as *integration by differentiation* [2]. Since the Gaussian pdf (2) has a unique
124  global maximum, its mean equals its mode (its maximum), which can be found by maximising
125  the log-pdf. Therefore, $\mu$ is obtained by solving $\partial_x \ln p(x|\mu, \mathbf{C}) = 0$:

$$-\partial_x \ln p(x|\mu, \mathbf{C})\Big|_{x_{\max}} = \partial_x \left[\frac{1}{2}(x - \mu)^{\mathsf{T}}\mathbf{C}^{-1}(x - \mu)\right]\Big|_{x_{\max}} = \mathbf{C}^{-1}(x - \mu)\big|_{x_{\max}} = 0, \tag{3}$$

126  which gives $x_{\max} = \mu$. Similarly, it is straightforward to verify that $\partial_x \partial_{x^{\mathsf{T}}} \ln p(x|\mu, \mathbf{C}) = -\mathbf{C}^{-1}$.
127  Therefore, $\mathbf{C}$ is found by $\mathbf{C} = -[\partial_x \partial_{x^{\mathsf{T}}} \ln p(x|\mu, \mathbf{C})]^{-1}$, i.e., by identifying the coefficient matrix
128  of the quadratic term in the exponent, inverting, and multiplying by $-1$.

129      **Generating samples of GRFs.** By virtue of the spectral theorem, any covariance matrix
130  $\mathbf{C}$ admits at least one "matrix square root" $\sqrt{\mathbf{C}}$ such that $\sqrt{\mathbf{C}}^2 = \mathbf{C}$. The usual procedure to
131  generate a GRF $x$ is to draw a "white noise" vector $\xi$ (namely, a vector of uncorrelated, zero-
132  mean, unit-variance Gaussian variables). We then compute $x = \sqrt{\mathbf{C}}\xi + \mu$ for any "matrix
133  square root" $\sqrt{\mathbf{C}}$ of $\mathbf{C}$. Then $x$ is a sample from the desired pdf given by Equation (2).

134      **Marginals and conditionals of GRFs.** Consider a "joint" GRF $\begin{pmatrix} x \\ y \end{pmatrix}$ with mean $\mu = \begin{pmatrix} \mu_x \\ \mu_y \end{pmatrix}$

135  and covariance $\mathbf{C} = \begin{pmatrix} \mathbf{C}_{xx} & \mathbf{C}_{xy} \\ \mathbf{C}_{yx} & \mathbf{C}_{yy} \end{pmatrix}$. A standard result is that the marginal pdf of $x$ and the

---

[2]In these notes, we work with real GRFs for simplicity, but the treatment could be generalised to complex GRFs.

[3]It can be verified that the mean $\langle x \rangle = \int_{-\infty}^{+\infty} x\, p(x|\mu, \mathbf{C})$ equals $\mu$ and the covariance matrix $\langle(x - \mu)(x - \mu)^{\mathsf{T}}\rangle = \int_{-\infty}^{+\infty}(x - \mu)(x - \mu)^{\mathsf{T}} p(x|\mu, \mathbf{C})$ equals $\mathbf{C}$ by computing the Gaussian integrals.

conditional pdf of $x$ given $y$ (i.e. $p(x)$ and $p(x|y)$, respectively) are also Gaussian, with means and covariances given below. For the marginals,

$$\langle x \rangle_{p(x)} = \int x\, p(x)\, \mathrm{d}x = \iint x\, p(x,y)\, \mathrm{d}x\, \mathrm{d}y = \mu_x, \tag{4}$$

$$\langle (x - \mu_x)(x - \mu_x)^\mathsf{T} \rangle_{p(x)} = \iint (x - \mu_x)(x - \mu_x)^\mathsf{T}\, p(x,y)\, \mathrm{d}x\, \mathrm{d}y = \mathbf{C}_{xx}, \tag{5}$$

i.e. the marginal mean and covariance are the corresponding blocks of the joint mean and covariance. Less trivially,

$$\langle x \rangle_{p(x|y)} \equiv \mu_{x|y} \qquad \text{where} \qquad \mu_{x|y} = \mu_x + \mathbf{C}_{xy}\mathbf{C}_{yy}^{-1}(y - \mu_y), \tag{6}$$

$$\langle (x - \mu_{x|y})(x - \mu_{x|y})^\mathsf{T} \rangle_{p(x|y)} \equiv \mathbf{C}_{x|y} \qquad \text{where} \qquad \mathbf{C}_{x|y} = \mathbf{C}_{xx} - \mathbf{C}_{xy}\mathbf{C}_{yy}^{-1}\mathbf{C}_{yx}. \tag{7}$$

The proof uses the notion of characteristic function of pdfs [see e.g. 3, appendix A].

## 2.2 A linear field-level model

In field-level inference tasks, the problem involves estimating an unknown signal field $s$ from noisy, incomplete observations $d$. A common linear data model is $d = s + n$, where $\begin{pmatrix} s \\ d \end{pmatrix}$ is assumed to form a joint GRF.[4] Both the signal $s$ and the additive noise $n$ are assumed to be GRFs with known covariance properties. The signal covariance matrix is denoted $\mathbf{C}_{ss} \equiv \mathbf{S} = \langle ss^\mathsf{T} \rangle$. The noise $n$ is assumed to have zero mean, a noise covariance matrix $\mathbf{C}_{nn} \equiv \mathbf{N} = \langle nn^\mathsf{T} \rangle$, and to be uncorrelated with the signal ($\mathbf{C}_{sn} = \langle sn^\mathsf{T} \rangle = 0$; $\mathbf{C}_{ns} = \langle ns^\mathsf{T} \rangle = 0$). From this, we obtain:

$$\mathbf{C}_{dd} = \mathbf{S} + \mathbf{N}, \tag{8}$$

$$\mathbf{C}_{sd} = \mathbf{C}_{ss} + \mathbf{C}_{sn} = \mathbf{C}_{ss} = \mathbf{S}, \tag{9}$$

where $\mathbf{C}_{dd} = \langle dd^\mathsf{T} \rangle$ and $\mathbf{C}_{sd} = \langle sd^\mathsf{T} \rangle$.

## 2.3 The Wiener filter posterior

Using the theorem for the conditionals of GRFs given in Section 2.1, the posterior $p(s|d)$ for the signal $s$ given the data $d$ is a Gaussian pdf with mean $\mu_{s|d} = \mu_s + \mathbf{C}_{sd}\mathbf{C}_{dd}^{-1}(d - \mu_d)$ and covariance matrix $\mathbf{C}_{s|d} = \mathbf{C}_{ss} - \mathbf{C}_{sd}\mathbf{C}_{dd}^{-1}\mathbf{C}_{ds}$. Substituting Equations (8) and (9) yields the final expressions:

$$\mu_{s|d} = \mu_s + \mathbf{S}(\mathbf{S} + \mathbf{N})^{-1}(d - \mu_d) = \mu_s + (\mathbf{S}^{-1} + \mathbf{N}^{-1})^{-1}\mathbf{N}^{-1}(d - \mu_d), \tag{10}$$

$$\mathbf{C}_{s|d} = \mathbf{S} - \mathbf{S}(\mathbf{S} + \mathbf{N})^{-1}\mathbf{S} = (\mathbf{S}^{-1} + \mathbf{N}^{-1})^{-1} = \sqrt{\mathbf{S}}\left(I + \sqrt{\mathbf{S}}\,\mathbf{N}^{-1}\,\sqrt{\mathbf{S}}\right)^{-1}\sqrt{\mathbf{S}}. \tag{11}$$

The various forms given in Equation (11) are mathematically equivalent. They can be derived from the Woodbury matrix identity: for any conformable matrices $\mathbf{A}$, $\mathbf{U}$, $\mathbf{B}$ and $\mathbf{V}$, one has $(\mathbf{A} + \mathbf{U}\mathbf{B}\mathbf{V})^{-1} = \mathbf{A}^{-1} - \mathbf{A}^{-1}\mathbf{U}\left(\mathbf{B}^{-1} + \mathbf{V}\mathbf{A}^{-1}\mathbf{U}\right)^{-1}\mathbf{V}\mathbf{A}^{-1}$. Therefore, with $\mathbf{A} = \mathbf{S}^{-1}$, $\mathbf{B} = \mathbf{N}^{-1}$, $\mathbf{U} = \mathbf{V} = \mathbf{I}$, we obtain: $(\mathbf{S}^{-1} + \mathbf{N}^{-1})^{-1} = \mathbf{S} - \mathbf{S}(\mathbf{S} + \mathbf{N})^{-1}\mathbf{S}$. With $\mathbf{A} = \mathbf{I}$, $\mathbf{B} = \mathbf{N}^{-1}$, $\mathbf{U} = \mathbf{V} = \sqrt{\mathbf{S}}$, we obtain: $\mathbf{I} - \sqrt{\mathbf{S}}(\mathbf{S} + \mathbf{N})^{-1}\sqrt{\mathbf{S}} = (\mathbf{I} + \sqrt{\mathbf{S}}\mathbf{N}^{-1}\sqrt{\mathbf{S}})^{-1}$. Hence, $\sqrt{\mathbf{S}}\left(\mathbf{I} + \sqrt{\mathbf{S}}\,\mathbf{N}^{-1}\,\sqrt{\mathbf{S}}\right)^{-1}\sqrt{\mathbf{S}} = \mathbf{S} - \mathbf{S}(\mathbf{S} + \mathbf{N})^{-1}\mathbf{S}$. In practice, it is always useful to check which of these forms yields the most numerically stable results, especially when the matrices have high condition numbers.

---

[4] All of the equations given in this section can be generalised to a linear data model of the form $d = \mathbf{A}s + n$ where $\mathbf{A}$ is a fixed matrix, and to complex fields.

162   Equations (10) and (11) are known as the *Wiener filter* equations. Performing Wiener
163 filtering (also referred to as "reconstructing the signal" or "denoising the data") consists in
164 computing the Wiener covariance matrix $\mathbf{C}_{s|d}$ and the Wiener filter mean field $\boldsymbol{\mu}_{s|d}$. The
165 mean of the reconstruction corresponds to the maximum a posteriori (MAP) estimator. Since
166 all pdfs are Gaussian, it is straightforward to verify that this estimator is also the optimal linear
167 estimator for signal reconstruction, in the sense of minimising the mean-squared error.

168   In addition to providing the point estimate $\boldsymbol{\mu}_{s|d}$, Wiener filtering enables the characteri-
169 sation of the full posterior distribution of the signal given the data. Using $\boldsymbol{\mu}_{s|d}$ and $\mathbf{C}_{s|d}$, it is
170 straightforward to draw constrained realisations of the signal given the data, i.e., samples of
171 the posterior $p(s|d)$, using the procedure given in Section 2.1: for any white-noise vector $\boldsymbol{\xi}$
172 of appropriate dimension, $\tilde{s} \equiv \boldsymbol{\mu}_{s|d} + \sqrt{\mathbf{C}_{s|d}}\boldsymbol{\xi}$ is a sample of $p(s|d)$. For uncorrelated draws
173 of $\boldsymbol{\xi}$, these $\tilde{s}$ samples will also be uncorrelated. The empirical mean and covariance of the
174 samples obtained in this fashion will approximate the Wiener filter mean $\boldsymbol{\mu}_{s|d}$ and covariance
175 $\mathbf{C}_{s|d}$, respectively.

176   **Derivation.** The Wiener filter equations can be derived from the general expressions for
177 conditionals of GRFs (Equations (6) and (7)). We provide an alternative derivation that di-
178 rectly relates to Bayes' theorem. The canonical expression for a Gaussian log-pdf is:

$$-2\ln p(\boldsymbol{x}|\boldsymbol{\mu}, \mathbf{C}) = \ln|2\pi\mathbf{C}| + (\boldsymbol{x} - \boldsymbol{\mu})^{\mathsf{T}}\mathbf{C}^{-1}(\boldsymbol{x} - \boldsymbol{\mu}) = \ln|2\pi\mathbf{C}| + \boldsymbol{\eta}^{\mathsf{T}}\boldsymbol{\Lambda}^{-1}\boldsymbol{\eta} - 2\boldsymbol{\eta}^{\mathsf{T}}\boldsymbol{x} + \boldsymbol{x}^{\mathsf{T}}\boldsymbol{\Lambda}\boldsymbol{x} \quad (12)$$

179 where $\boldsymbol{\Lambda} \equiv \mathbf{C}^{-1}$ (the *precision matrix*) and $\boldsymbol{\eta} \equiv \mathbf{C}^{-1}\boldsymbol{\mu}$. Assuming $\boldsymbol{\mu}_s = \boldsymbol{\mu}_d = 0$ (it is straightfor-
180 ward to reinstate the mean if it is non-zero), the log-prior and log-likelihood for the Wiener
181 filtering problem are:

$$-2\ln p(s) = \ln|2\pi\mathbf{S}| + s^{\mathsf{T}}\mathbf{S}^{-1}s, \quad (13)$$

$$-2\ln p(d|s) = \ln|2\pi\mathbf{N}| + (d - s)^{\mathsf{T}}\mathbf{N}^{-1}(d - s) = \ln|2\pi\mathbf{N}| + \boldsymbol{\eta}^{\mathsf{T}}\mathbf{N}\boldsymbol{\eta} - 2\boldsymbol{\eta}^{\mathsf{T}}s + s^{\mathsf{T}}\mathbf{N}^{-1}s, \quad (14)$$

182 with $\boldsymbol{\eta} \equiv \mathbf{N}^{-1}d$. Using Bayes' theorem (Equation (1)),

$$-2\ln p(s|d) = \text{const} - 2\boldsymbol{\eta}^{\mathsf{T}}s + s^{\mathsf{T}}(\mathbf{S}^{-1} + \mathbf{N}^{-1})s. \quad (15)$$

183 Therefore, the log-posterior takes the canonical form of a Gaussian log-pdf with covariance
184 matrix $\mathbf{W} = (\mathbf{S}^{-1} + \mathbf{N}^{-1})^{-1}$, and its mean is given by $\mathbf{W}\boldsymbol{\eta} = (\mathbf{S}^{-1} + \mathbf{N}^{-1})^{-1}\mathbf{N}^{-1}d$, which corre-
185 sponds to Equations (11) and (10).

## 186   2.4   Wiener filtering example

187 For illustrative purposes, let us consider the case of a two-dimensional periodic cosmological
188 signal field $s$, generated using the BBKS power spectrum [4]. The signal covariance matrix
189 $\mathbf{S}$ is diagonal in Fourier space. For the noise covariance matrix $\mathbf{N}$, we choose it to be diago-
190 nal in configuration space with three distinct regions: one with low noise (representing, e.g.,
191 multiple overlapping survey passes), one with higher noise (representing, e.g., only one sur-
192 vey pass), and one with infinite noise (representing a masked region[5]). The data model is
193 $d = s + n$.

194   Figure 1 illustrates Wiener filtering for this toy model. From left to right, the panels show:
195 the true underlying signal $s$, the simulated data $d$, the Wiener-filtered data (the posterior
196 mean) $\boldsymbol{\mu}_{s|d}$, and one constrained realisation of the signal (one sample of the posterior), $\tilde{s}$.

---

[5]Note that in Bayesian field-level inference, no special treatment is required for masked regions: they can simply
be treated as regions with infinite noise. When $\mathbf{N}$ is diagonal in configuration space and only $\mathbf{N}^{-1}$ is required to
evaluate the Wiener filter equations, the coefficients of $\mathbf{N}^{-1}$ can be set to zero in masked regions.

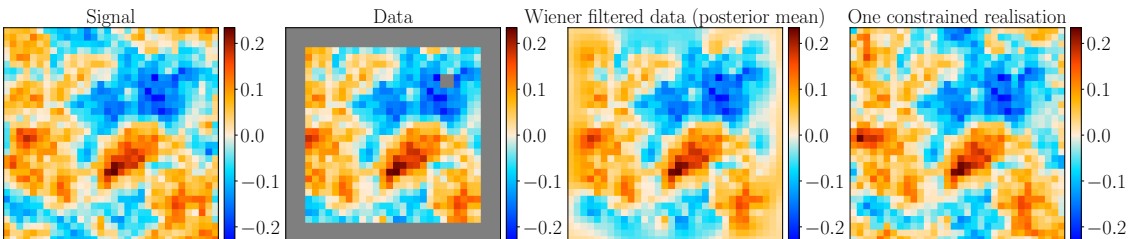

Figure 1: Illustration of Wiener filtering for a two-dimensional field-level inference problem with a linear model. From left to right, the panels show: the true underlying signal field, the data, the Wiener-filtered data (the posterior mean), and one constrained realisation of the signal.

## 3 Markov Chain Monte Carlo techniques for field-level inference

This section introduces the numerical sampling methods essential for exploring complex, high-dimensional posterior distributions encountered in field-level inference with non-linear models.

### 3.1 Overview of MCMC and the Metropolis-Hastings algorithm

Markov Chain Monte Carlo (MCMC) methods [e.g. 5] provide a powerful approach for sampling from probability distributions, particularly in high-dimensional spaces ($D \gtrsim 4$) where analytical solutions or direct visualisation become intractable. Markov chains are defined as sequences of states $\{x_i\}_{1 \le i \le n}$ that satisfy the *Markov property*: the information needed to predict the future is contained entirely in the present state of the process and does not depend on past states (i.e., the system has no "memory"). Mathematically, this is expressed in terms of conditional probabilities: the conditional probability distribution of future states, given past states and the present state, depends only on the present state and not on past states:

$$p(x_{n+1} | \{x_i\}_{1 \le i \le n}) = p(x_{n+1} | x_n). \tag{16}$$

**Metropolis-Hastings algorithm.** Markov chains are typically constructed using algorithms that propose moves which are either accepted or rejected. The simplest MCMC method is the Metropolis-Hastings (MH) algorithm. In this scheme, one first draws a candidate point $x^*$ from a proposal distribution $q(x^*|x)$ that depends on the current state $x = x_n$. One then evaluates the *Hastings ratio*,

$$r(x, x^*) \equiv \frac{p(x^*)}{p(x)} \frac{q(x|x^*)}{q(x^*|x)} \tag{17}$$

where $p(\cdot)$ is the (possibly unnormalised) target pdf. A single uniform draw $u \frown \mathcal{U}([0,1])$ determines the move: if $u < r(x, x^*)$, the chain jumps to $x_{n+1} = x^*$ (an acceptance), otherwise, it stays at $x_{n+1} = x$ (a rejection). The *acceptance rate* of the chain is defined as the ratio of the number of accepted moves to the total number of Metropolis-Hastings tests. Note that if $r(x, x^*) \ge 1$, the move is always accepted, and that it is important to *re-write* the value of $x$ in the chain in case of rejection. Under these assumptions, it is possible to prove that the chain has the target distribution as its stationary distribution, i.e., elements of the chain (asymptotically) become correlated samples of $p(\cdot)$.[6] In particular, after a *burn-in period*, the empirical averages of the chain converge to expectations under $p(\cdot)$.

---

[6]This is only a very simplified and practical guide to MCMC. A more extensive treatment requires introducing the notions *ergodicity* and *global/detailed balance*.

A common special case is that of a symmetric proposal distribution, which ensures that $q(\boldsymbol{x}^*|\boldsymbol{x}) = q(\boldsymbol{x}|\boldsymbol{x}^*)$. Then the Hastings ratio simplifies to

$$r(\boldsymbol{x}, \boldsymbol{x}^*) \equiv \frac{p(\boldsymbol{x}^*)}{p(\boldsymbol{x})}, \tag{18}$$

and such a move is called a *Metropolis update*. Unfortunately, theory offers no general prescription for designing the proposal distribution $q(\cdot)$. A typical choice is a Gaussian proposal distribution: in one dimension, $q(x^*|x) = \mathcal{G}(x, \sigma^2)$ for some "step size" $\sigma$; or in higher dimensions, $q(\boldsymbol{x}^*|\boldsymbol{x}) = \mathcal{G}(\boldsymbol{x}, \mathbf{C})$ for some proposal covariance matrix $\mathbf{C}$. The sampling efficiency depends crucially on the step size $\sigma$ or the proposal covariance matrix $\mathbf{C}$ (see Section 3.3). A frustrating property is that the optimal proposal distribution $q(\cdot)$ to sample from the target distribution $p(\cdot)$ is... the target distribution $p(\cdot)$ itself!

## 3.2 MCMC beyond Metropolis-Hastings

Field-level inference involves inherently high-dimensional parameter spaces, including the numerous pixel values representing cosmological fields. For such inference tasks, the MH algorithm will typically fail because the *curse of dimensionality* (i.e., the exponential increase in the volume of the parameter space with added dimensions) makes any random-walk proposal almost certainly land in regions of extremely low posterior probability, leading to a near-constant rejection of proposed states. This results in extremely poor chain mixing and convergence (see Section 3.3). Tuning effective proposal distributions for high-dimensional MH sampling is usually prohibitively difficult. In this section, more advanced MCMC techniques are discussed to overcome the limitations of the MH algorithm and improve sampling efficiency in cases where some information about the target pdf is available: conditionals (Gibbs sampling, Section 3.2.1) or gradients (Hamiltonian Monte Carlo, Section 3.2.2).

### 3.2.1 Gibbs Sampling

Gibbs sampling is an MCMC algorithm that generates samples from a joint probability distribution by iteratively sampling each variable from its conditional distribution given the current values of all other variables. By cycling through all variables (or blocks of variables), the procedure constructs a Markov chain whose stationary distribution is the desired joint distribution.

More specifically, let us denote by $p(\boldsymbol{x}, \boldsymbol{y})$ the target pdf ($\boldsymbol{x}$ and $\boldsymbol{y}$ can be vectors) and let us assume that we know how to generate samples from both the conditionals $p(\boldsymbol{x}|\boldsymbol{y})$ and $p(\boldsymbol{y}|\boldsymbol{x})$. Then Gibbs sampling is a special case of MH in which the proposal mechanism is to update $\boldsymbol{x}$ to $\boldsymbol{x}^*$ given $\boldsymbol{y}$, then update $\boldsymbol{y}$ to $\boldsymbol{y}^*$ given $\boldsymbol{x}$, and iterate. The proposal distributions are $q(\boldsymbol{x}^*|\boldsymbol{x}, \boldsymbol{y}) = p(\boldsymbol{x}^*|\boldsymbol{y})$ for updating $\boldsymbol{x}$ and $q(\boldsymbol{y}^*|\boldsymbol{x}, \boldsymbol{y}) = p(\boldsymbol{y}^*|\boldsymbol{x})$ for updating $\boldsymbol{y}$. Therefore, it is straightforward to verify that the Hastings ratio is always unity:

$$r(\boldsymbol{x}, \boldsymbol{x}^*|\boldsymbol{y}) = \frac{p(\boldsymbol{x}^*, \boldsymbol{y})}{p(\boldsymbol{x}, \boldsymbol{y})} \frac{q(\boldsymbol{x}|\boldsymbol{x}^*, \boldsymbol{y})}{q(\boldsymbol{x}^*|\boldsymbol{x}, \boldsymbol{y})} = \frac{p(\boldsymbol{x}^*|\boldsymbol{y})p(\boldsymbol{y})}{p(\boldsymbol{x}|\boldsymbol{y})p(\boldsymbol{y})} \frac{p(\boldsymbol{x}|\boldsymbol{y})}{p(\boldsymbol{x}^*|\boldsymbol{y})} = 1, \tag{19}$$

and similarly for $r(\boldsymbol{y}, \boldsymbol{y}^*|\boldsymbol{x})$.

As a rejection-free sampling technique, Gibbs sampling is particularly effective when the conditionals of the target pdf are known and easy to sample from. However, a limitation of Gibbs sampling arises when $\boldsymbol{x}$ and $\boldsymbol{y}$ are strongly degenerate: since the algorithm takes steps along the $\boldsymbol{x}$ and $\boldsymbol{y}$ directions in parameter space, it will be inefficient at generating samples from thin distributions that are not aligned with these directions.

### 3.2.2  Hamiltonian Monte Carlo

Hamiltonian Monte Carlo (HMC) [see 6], originally introduced as Hybrid Monte Carlo [7], is a gradient-based sampler that exploits Hamiltonian dynamics to propose samples, thereby enabling the exploration of complex, high-dimensional posteriors. The central idea is to use classical mechanics to solve statistical problems.

In HMC, we interpret the negative target log-pdf as a physical potential, $\psi(\boldsymbol{x}) \equiv -\ln p(\boldsymbol{x})$. We introduce a conjugate momentum $\boldsymbol{p}_i$ for each of the target variables $\boldsymbol{x}_i$ (the "positions"), as well as a mass matrix $\mathbf{M}$, to define a Hamiltonian:

$$H(\boldsymbol{x}, \boldsymbol{p}) \equiv \frac{1}{2} \boldsymbol{p}^{\intercal} \mathbf{M}^{-1} \boldsymbol{p} + \psi(\boldsymbol{x}). \tag{20}$$

Then the proposal mechanism is to move the system from the state $(\boldsymbol{x}, \boldsymbol{p})$ to a new state $(\boldsymbol{x}^*, \boldsymbol{p}^*)$ by following Hamilton's equations in parameter space for some amount of (fictitious) time $t$,

$$\frac{\mathrm{d}\boldsymbol{x}}{\mathrm{d}t} = \frac{\partial H}{\partial \boldsymbol{p}} = \mathbf{M}^{-1} \boldsymbol{p}, \tag{21}$$

$$\frac{\mathrm{d}\boldsymbol{p}}{\mathrm{d}t} = -\frac{\partial H}{\partial \boldsymbol{x}} = -\frac{\mathrm{d}\psi(\boldsymbol{x})}{\mathrm{d}\boldsymbol{x}}. \tag{22}$$

It can be shown that the Hastings ratio for such a proposal is

$$r(\boldsymbol{x}, \boldsymbol{x}^*) = \exp\left\{-[H(\boldsymbol{x}^*, \boldsymbol{p}^*) - H(\boldsymbol{x}, \boldsymbol{p})]\right\}. \tag{23}$$

Therefore the Hastings ratio is always unity if the Hamiltonian (the energy) of the system is exactly conserved when moving from $(\boldsymbol{x}, \boldsymbol{p})$ to $(\boldsymbol{x}^*, \boldsymbol{p}^*)$. Importantly, this statement is independent of the dimensionality of the problem. In practice, the Hamiltonian is never exactly conserved due to numerical errors; therefore an MH acceptance test should always be introduced to ensure convergence to the target pdf. After each MH test, the system is "kicked", meaning that the old momentum $\boldsymbol{p}$ is replaced by a new momentum vector drawn from its prior pdf $q(\boldsymbol{p})$, a Gaussian distribution with zero mean and covariance matrix $\mathbf{M}$ (this ensures that the algorithm will asymptotically explore all possible energy levels). Importantly, Equation (22) involves the gradient of the potential, $\mathrm{d}\psi(\boldsymbol{x})/\mathrm{d}\boldsymbol{x}$. Access to the gradient of the target log-pdf is a crucial requirement underlying HMC.

Practical implementations of HMC require discretising the equations of motion (Equations (21) and (22)). The usual choice, taken from the literature on computer simulations using particles, is the *leapfrog* (or *kick-drift-kick*) integrator. Given a step size $\epsilon$, the system is moved from $(\boldsymbol{x}, \boldsymbol{p})$ at time $t$ to $(\boldsymbol{x}^*, \boldsymbol{p}^*)$ at time $t + \epsilon \times N_{\text{steps}}$ by a sequence of $N_{\text{steps}}$ kick-drift-kick operations, each given by

$$\boldsymbol{p}_i\left(t + \frac{\epsilon}{2}\right) = \boldsymbol{p}_i(t) - \frac{\epsilon}{2} \left.\frac{\partial \psi}{\partial \boldsymbol{x}_i}\right|_{\boldsymbol{x}_i(t)} \qquad \text{(kick)}, \tag{24}$$

$$\boldsymbol{x}_i(t + \epsilon) = \boldsymbol{x}_i(t) + \frac{\epsilon}{m_i} \boldsymbol{p}_i\left(t + \frac{\epsilon}{2}\right) \qquad \text{(drift)}, \tag{25}$$

$$\boldsymbol{p}_i(t + \epsilon) = \boldsymbol{p}_i\left(t + \frac{\epsilon}{2}\right) - \frac{\epsilon}{2} \left.\frac{\partial \psi}{\partial \boldsymbol{x}_i}\right|_{\boldsymbol{x}_i(t+\epsilon)} \quad \text{(kick)}, \tag{26}$$

where $i$ labels one of the parameters, and we have assumed (a very common choice) that the mass matrix $\mathbf{M}$ is diagonal with elements $\{m_i\}$. The leapfrog integrator is a second-order integrator and is often used in HMC because it is *symplectic*, i.e., it conserves energy; this property is crucial for maintaining a high acceptance rate, which is needed to explore parameter space. An HMC sampler with a leapfrog integrator involves several free parameters that replace the

step size $\sigma$ of the MH algorithm, in particular: the mass matrix $\mathbf{M}$ (restricted to the mass $m_i$ of each parameter in the case of a diagonal mass matrix), the step size of the leapfrog integrator $\epsilon$, and the number of leapfrog steps between two MH tests $N_{\mathrm{steps}}$.

To this day, the core idea that underlies HMC (using classical mechanics to build a proposal distribution for MCMC) remains the only known way to sample the high-dimensional (multi-hundred to multi-million) pdfs that appear in field-level inference with non-linear models. HMC beats the curse of dimensionality by (i) using the conservation of the Hamiltonian and (ii) exploiting gradients. Point (i) ensures that the acceptance rate remains high, whereas any random-walk proposal distribution within a vanilla MH algorithm would yield an acceptance rate of zero. Point (ii) is an important requirement: easy access to the gradient of the target log-pdf (and therefore of the data model) explains the rapid rise in popularity of libraries such as `JAX` or `PyTorch`, which implement the very powerful technology of *automatic differentiation* (or "autodiff"). In recent years, some variants of HMC, such as the No-U-Turn Sampler [NUTS, 8], have been introduced and used for field-level inference, but the core idea remains unchanged.

## 3.3 Diagnostics of Markov chains

### 3.3.1 Trace plots, burn-in, and mixing

In Bayesian statistics, assessing the quality (convergence and efficiency) of Markov chains is crucial to ensuring reliable inference results, and this is particularly true for field-level inference. The most common tool to visually inspect the behaviour of Markov chains is *trace plots* (i.e., plots of the value of a sampled parameter as a function of position in the chain; see Figure 2).

**Burn-in and mixing.** Trace plots typically highlight the existence of a "burn-in" (or "warm-up") phase, during which the chain remains correlated with its starting point before reaching the high-probability regions of the target pdf. These initial samples should be discarded as the chain has not yet reached its stationary distribution. When possible, starting several chains from different initial points provides a better visual assessment of the length of the burn-in phase. Trace plots of different chains can also be used to assess their mixing, indicating how well the chains explore parameter space.

**Tuning the proposal distribution.** Along with the fraction of accepted samples, trace plots also provide immediate visual diagnostics of sampling efficiency and clues to tuning the proposal distribution. A "healthy" Markov chain (Figure 2, central panel) should scatter in parameter space without long, visible correlations between samples. Such behaviour usually lies in the sweet spot between two opposite undesirable behaviours: (i) slow drifting of the chain in parameter space (indicating acceptance of many small moves; see Figure 2, left-hand panel) and a very high acceptance rate (usually $\gtrsim$ 90% or higher); and (ii) long plateaus in the trace plot (indicating rejections; see Figure 2, right-hand panel) and a low acceptance rate (usually $\lesssim$ 5%). Behaviour (i) indicates that the proposal distribution is too narrow for the parameter under consideration: the step size $\sigma$ is too small for MH, or, for HMC, $\epsilon$ and/or $N_{\mathrm{steps}}$ are too small, or the parameter is too "heavy" to be moved. On the contrary, behaviour (ii) indicates that the proposal distribution is too broad for the parameter under consideration: the step size $\sigma$ is too large for MH, or, for HMC, $\epsilon$ and/or $N_{\mathrm{steps}}$ are too large, or the parameter is too "light".

### 3.3.2 Convergence assessment: the Gelman-Rubin test

The well-known Gelman-Rubin test [see 9] is a quantitative diagnostic for assessing the convergence of multiple MCMC chains. Given $m$ independent chains started at different points

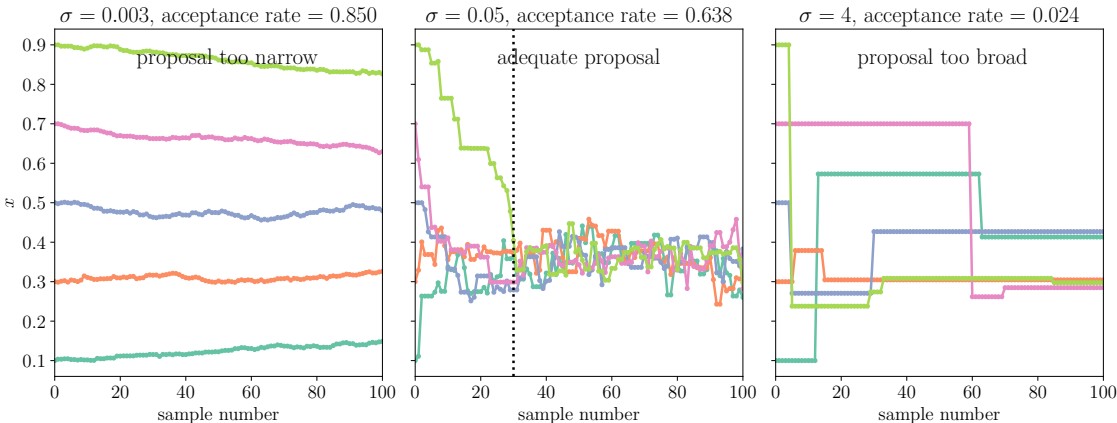

Figure 2: Diagnostics of Markov chains using trace plots. The central panel shows the typical behaviour of a "healthy" Markov chain with an adequate proposal distribution, and allows assessment of the length of the burn-in period (here, about 30 samples). The left-hand panel shows the typical behaviour of a Markov chain with too narrow a proposal distribution, and the right-hand panel shows the typical behaviour of a Markov chain with too broad a proposal distribution.

in parameter space, each of length $n$ and with target pdf $p(\cdot)$, we introduce the "between-chain" variance $B(x)$ and the "within-chain" variance $W(x)$ for parameter $x$, defined, using the samples $\left\{x_{ij}\right\}_{1\leq i\leq n,\, 1\leq j\leq m}$ by

$$B(x) \equiv \frac{n}{m-1}\sum_{j=1}^{m}\left(\bar{x}_{\cdot j}-\bar{x}_{\cdot\cdot}\right)^2, \tag{27}$$

$$W(x) \equiv \frac{1}{m}\sum_{j=1}^{m}s_j^2, \tag{28}$$

where

$$\bar{x}_{\cdot j} \equiv \sum_{i=1}^{n}x_{ij}, \quad \bar{x}_{\cdot\cdot} \equiv \frac{1}{m}\sum_{j=1}^{m}\bar{x}_{\cdot j}, \quad s_j^2 \equiv \frac{1}{n-1}\sum_{i=1}^{n}\left(x_{ij}-\bar{x}_{\cdot j}\right)^2. \tag{29}$$

We can then define two estimators of the posterior variance for each parameter:

$$\widehat{\mathrm{var}}^{-}(x) \equiv W(x), \tag{30}$$

$$\widehat{\mathrm{var}}^{+}(x) \equiv \frac{n}{n-1}W(x)+\frac{1}{n}B(x). \tag{31}$$

It can be shown that $\widehat{\mathrm{var}}^{-}(x)$ always underestimates the marginal variance of $x$ under $p(\cdot)$, that $\widehat{\mathrm{var}}^{+}(x)$ always overestimates this variance, and that the two estimators converge asymptotically to the marginal variance of $x$.

Thus, the potential scale reduction factor (PSRF), or $\hat{R}$ statistic, defined by

$$\hat{R}(x) \equiv \sqrt{\frac{\widehat{\mathrm{var}}^{+}(x)}{\widehat{\mathrm{var}}^{-}(x)}}, \tag{32}$$

should tend to unity. The Gelman-Rubin test consists in checking that $\hat{R}\to 1$ as $n\to\infty$ for each parameter. Typically, we aim for $\hat{R}(x)-1\lesssim 10^{-2}$ for all parameters $x$ (although the threshold for deciding that convergence is achieved is arbitrary).

### 3.3.3   Correlation length and effective sample size

Correlation length and effective sample size [10] summarise the autocorrelation structure of an MCMC chain, revealing how many practically independent draws are available for reliable uncertainty estimation. The basic idea is that the samples in the Markov chain are not independent, and we must estimate the effective number of independent samples.

**Correlation length.** For a Markov chain, the *integrated autocorrelation length* is defined as $\tau_x \equiv \sum_{\tau=-\infty}^{+\infty} \rho_x(\tau)$, where $\rho_x(\tau)$ is the normalised *autocorrelation function* of the stochastic process that generated the chain for $x$. We can estimate $\rho_x(\tau)$ from a chain $\{x_i\}_{1 \leq i \leq n}$ of finite length $n$ via $\hat{\rho}_x(\tau) = \hat{c}_x(\tau)/\hat{c}_x(0)$, where

$$\hat{c}_x(\tau) \equiv \frac{1}{n-\tau} \sum_{i=1}^{n-\tau} (x_i - \mu_x)(x_{i+\tau} - \mu_x) \quad \text{and} \quad \mu_x \equiv \frac{1}{n} \sum_{i=1}^{n} x_i. \tag{33}$$

In practice, it is more computationally efficient to compute $\hat{c}_x(\tau)$ using fast Fourier transforms than to sum it directly. The Wiener-Khinchin theorem states that the autocorrelation function of a signal is the inverse Fourier transform of its power spectral density:

$$\hat{c}_x(\tau) \propto \mathcal{F}^{-1}\left\{ |\mathcal{F}\{x - \mu_x\}|^2 \right\}. \tag{34}$$

A good estimator of the integrated autocorrelation length $\tau_x$ is:

$$\hat{\tau}_x = 1 + 2 \sum_{\tau=1}^{m} \hat{\rho}_x(\tau) \tag{35}$$

for some $m \ll n$ (an automated windowing procedure can be used).[7]

**Effective sample size.** The estimated effective sample size (ESS) $n_{\text{eff}}$ for the variable $x$ is the total length of the chain divided by the estimated integrated autocorrelation length, i.e., $n_{\text{eff}} \equiv n/\hat{\tau}_x$. Since the integrated autocorrelation length directly quantifies the Monte Carlo error, the ESS for a given computational budget characterises the efficiency of the MCMC sampler. In particular, the figure of merit for the sampling efficiency of any MCMC algorithm is the *ESS per function* (data model, likelihood, gradient) *evaluation*.

## 4   Field-level inference with non-linear models

This section extends field-level inference to non-linear cosmological models, which requires the advanced MCMC sampling techniques described in Section 3.2.

### 4.1   A non-linear field-level model with primordial non-Gaussianity

Unlike the linear model introduced in Section 2.2, non-linear cosmological data models describe scenarios in which fields, such as the density contrast $\delta$, are no longer Gaussian random fields. Let us extend our toy linear data model (Section 2.4) to this case.

The signal $s$ that we seek to reconstruct is now a white-noise field (in configuration space). The primordial gravitational potential $\Phi_{\text{L}}$ is a Gaussian random field with phases inherited from $s$, zero mean, and a "primordial" power spectrum given by $P_{\text{L}}(k) = A_s k^{n_s - 1}$ (i.e., a diagonal covariance matrix in Fourier space), where $A_s$ and $n_s$ are cosmological parameters. The non-linear gravitational potential $\Phi_{\text{NL}}$ is given by

$$\Phi_{\text{NL}} = \Phi_{\text{L}} + f_{\text{NL}} \Phi_{\text{L}}^2, \tag{36}$$

---

[7]See, e.g., the emcee documentation at https://emcee.readthedocs.io/en/stable/tutorials/autocorr/ for further discussion and a Python implementation.

where $f_{\mathrm{NL}}$ is another cosmological parameter.[8] The density contrast field $\delta$ is obtained by applying a transfer function (in Fourier space),

$$\delta(\boldsymbol{k}) = D_1 T(k) \Phi_{\mathrm{NL}}(\boldsymbol{k}), \tag{37}$$

where $D_1$ is a fixed coefficient (the cosmological growth factor) and $T(\cdot)$ is the BBKS transfer function.

For the observational part of our data model, we retain the assumptions made in Section 2.4. Namely, we assume that the data $\boldsymbol{d}$ are a noisy observation of the density contrast $\boldsymbol{\delta}$, given by $\boldsymbol{d} = \boldsymbol{\delta}(\boldsymbol{s}) + \boldsymbol{n}$, where $\boldsymbol{n}$ is additive Gaussian noise with zero mean and a covariance matrix $\mathbf{N}$ that is diagonal in configuration space. Thus, when $f_{\mathrm{NL}} = 0$, our model is equivalent to the linear model described in Section 2.4, in which the density contrast $\boldsymbol{\delta}$ serves as the signal to be reconstructed, and the BBKS power spectrum is $P(k) = D_1^2 T(k)^2 P_{\mathrm{L}}(k)$.

Altogether, this model is a Bayesian hierarchical field-level model (see Section 1.4) for the signal and cosmological parameters, given the data.

## 4.2 Constructing the posterior distribution and its gradient

Contrary to Wiener filtering, for which an analytic expression for the posterior distribution is available, in field-level inference with non-linear models we generally do not have an analytic expression for the posterior. Therefore, we must explicitly specify the (unnormalised) log-prior, log-likelihood, and log-posterior of the problem, and rely on MCMC techniques to sample the posterior distribution. The process involves defining the natural prior on the signal and the likelihood for the observed data, given the signal and parameters.

Let us write down the log-posterior distribution for our non-linear field-level model described in Section 4.1. We first assume that cosmological parameters $\{A_{\mathrm{s}}, n_{\mathrm{s}}, f_{\mathrm{NL}}\}$ are fixed, so that we sample only the signal field $\boldsymbol{s}$. Since the signal $\boldsymbol{s}$ is white noise, the natural prior distribution is a Gaussian with zero mean and identity covariance matrix:

$$\ln p(\boldsymbol{s}) = -\frac{1}{2} \boldsymbol{s}^{\intercal} \boldsymbol{s} + \mathrm{const}. \tag{38}$$

The log-likelihood captures our assumptions about the observational process. Given the assumptions made about the noise distribution, the probability of the data $\boldsymbol{d}$ given the signal $\boldsymbol{s}$ is a Gaussian distribution centred on the data model prediction $\boldsymbol{\delta}(\boldsymbol{s})$, with $\mathbf{N}$ as covariance matrix. Therefore,

$$\ln p(\boldsymbol{d}|\boldsymbol{s}) = -\frac{1}{2} [\boldsymbol{\delta}(\boldsymbol{s}) - \boldsymbol{d}]^{\intercal} \mathbf{N}^{-1} [\boldsymbol{\delta}(\boldsymbol{s}) - \boldsymbol{d}] + \mathrm{const}. \tag{39}$$

Using Bayes' theorem, as given in Equation (1), we obtain the final form of the log-posterior and the corresponding potential,

$$\psi(\boldsymbol{s}) = -\ln p(\boldsymbol{s}|\boldsymbol{d}) = \frac{1}{2} \boldsymbol{s}^{\intercal} \boldsymbol{s} + \frac{1}{2} [\boldsymbol{\delta}(\boldsymbol{s}) - \boldsymbol{d}]^{\intercal} \mathbf{N}^{-1} [\boldsymbol{\delta}(\boldsymbol{s}) - \boldsymbol{d}] + \mathrm{const}. \tag{40}$$

As discussed in Section 3.2.2, for use with a gradient-based sampler, we require the gradient of the potential with respect to the signal $\boldsymbol{s}$, $\mathrm{d}\psi(\boldsymbol{s})/\mathrm{d}\boldsymbol{s}$. The term corresponding to the log-prior part is trivially differentiated; however, the term corresponding to the log-likelihood requires, when applying the chain rule, the gradient of the data model with respect to the signal ($\mathrm{d}\boldsymbol{\delta}(\boldsymbol{s})/\mathrm{d}\boldsymbol{s}$ for our model). For this reason, the historical approach to field-level inference has been to compute analytically and then write "by hand" computer functions that evaluate

---

[8]Such a model is known as local-type primordial non-Gaussianity, and $f_{\mathrm{NL}}$ is related to the physics of cosmological inflation.

the gradients of data models (including $N$-body simulations). Such an approach, although tedious, remains partly in use because it allows full control over the gradient code and, in particular, its memory footprint. In recent years, the powerful technique of automatic differentiation has emerged and been leveraged for field-level inference. With this approach, the log-prior, log-likelihood, and log-posterior functions are implemented in a framework such as JAX, enabling automatic differentiation to compute the gradients essential for sampling. Comparing the gradient code with finite differencing is a useful sanity check, and essential when gradients are written manually. To do so, at a fiducial point in parameter space, we can plot the gradient $\mathrm{d}\psi(s)/\mathrm{d}s$ against the pixel/voxel index in $s$, computed either from the (manual or automatic) gradient code or via finite differences of the potential $\psi(s)$ itself. The two predictions should agree to a degree typically close to machine precision.

### 4.3 Sampling with a gradient-based sampler

As discussed in Section 3.2, high-dimensional Bayesian sampling problems, such as those encountered in field-level inference with non-linear models, require sampling strategies beyond the MH algorithm. In particular, exploration of the high-dimensional posterior distribution of the signal field $s$ requires a gradient-based sampler: HMC or variants such as NUTS. Some field-level inference codes implement their own sampler to maintain full control over the sampling strategy and its efficiency. Other applications employ libraries such as Stan, PyMC, NumPyro, or BlackJAX, which provide off-the-shelf gradient-based samplers, some of which allow auto-tuning of the proposal distribution.

Using a gradient-based sampler, we obtain samples of the posterior for the signal, $p(s|d)$. As natural by-products, we also obtain samples of all the latent fields appearing in the problem ($\boldsymbol{\Phi}_\mathrm{L}$, $\boldsymbol{\Phi}_\mathrm{NL}$, and $\boldsymbol{\delta}$ for our model in Section 4.1).

### 4.4 Chain diagnostics and posterior summaries

Field-level inference usually requires a phase of tuning the proposal distribution used by the sampler, in particular the leapfrog step size $\epsilon$, the number of steps $N_\mathrm{steps}$, and the mass matrix. This can be done by following the principles given in Section 3.3.1. Once this is done, the proposal distribution should be held fixed before starting to record the actual Markov chain. We can then assess the length of any residual burn-in period to discard. To do so, *trace plots* can be used; another useful plot is of the value of the *negative log-likelihood* (or log-posterior) *as a function of the sample index* (it is often recorded by the sampler; otherwise it can be re-evaluated). The usual behaviour of $-\ln p(d|s)$ along the chain is to start at a high value, drop, and then oscillate around a constant value after burnt-in.

Starting the chain from an *overdispersed state* is good practice in field-level inference, as it provides useful burn-in diagnostics. This means initialising the sampler far from the expected high-density posterior regions; for example, when sampling a white-noise field $s$, we can start the chain from any white-noise field scaled by a factor of $10^{-3}$. Such an initialisation allows one to check the *sequential posterior power spectrum* of samples of the reconstructed signal or of any latent field appearing in the problem, such as the density contrast $\boldsymbol{\delta}$ (see Figure 3). The sequential power spectrum of samples gives the length of the burn-in phase as a function of scale and permits checking that the power spectrum of the fields converges to the expected value (either the ground truth or a theoretical prediction). It is also a powerful diagnostic for potential model misspecification when working with real data.

Following sampling, standard MCMC diagnostics (see Sections 3.3.2 and 3.3.3) can be performed to assess convergence and mixing. Computing the ESS is useful for gauging the number of independent samples. Although chains are rarely converged in the Gelman-Rubin sense, it is always possible to report the PSRF, $\hat{R}$, at the field level.

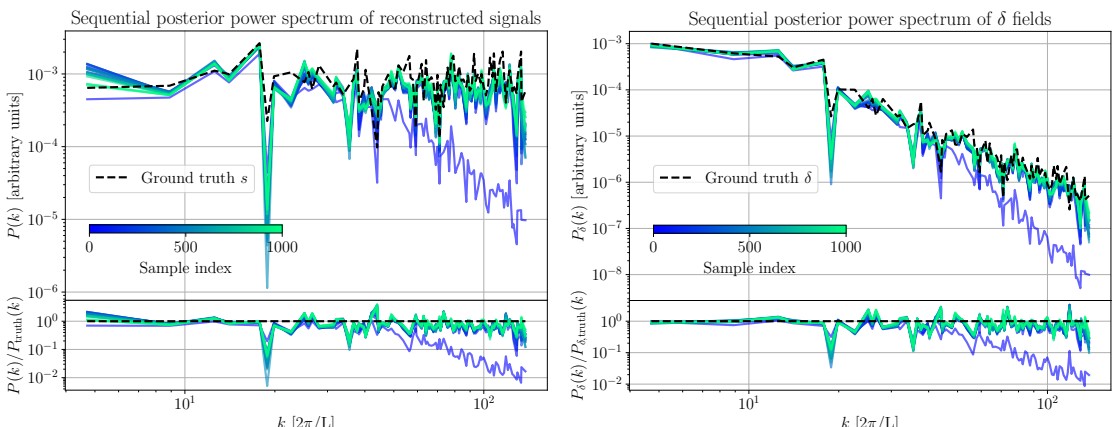

Figure 3: Sequential posterior power spectrum of the white noise signal field $s$ (left-hand panel) and the density contrast field $\delta$ (right-hand panel) as a function of the sample index. The dashed line denotes the power spectrum of the corresponding ground-truth fields.

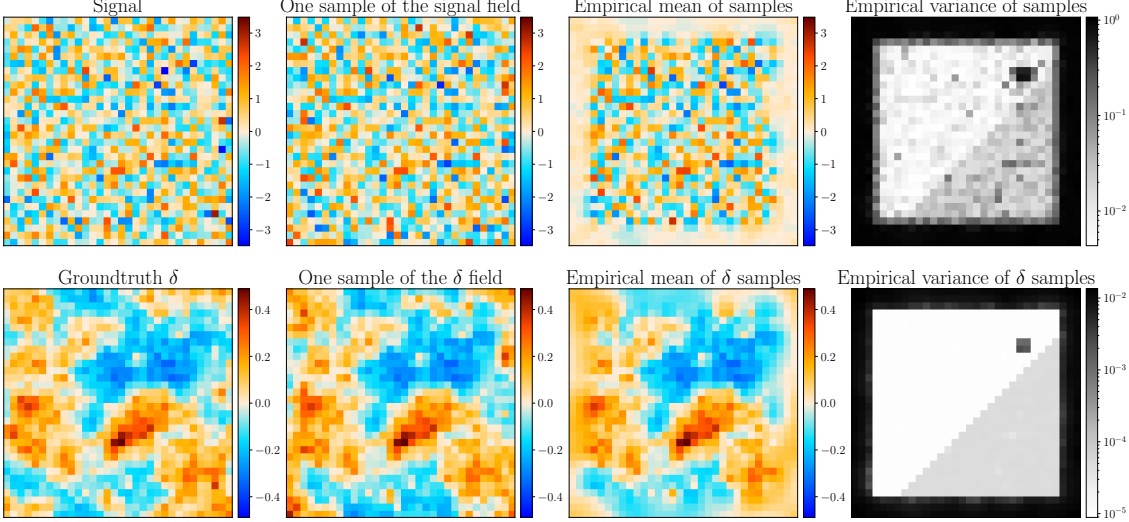

Figure 4: Posterior summaries for a non-linear field-level inference problem. The first row shows, from left to right, the ground-truth signal $s$ to be inferred, one sample, the empirical mean, and the empirical variance of the samples. The second row shows analogous fields for a latent field appearing in the problem: the density contrast $\delta$.

The full set of samples obtained is the actual result of field-level inference, with the crucial property that their density is proportional to the posterior density. However, for communication purposes it is usually necessary to plot empirical posterior summaries (see Figure 4). We usually visualise the empirical mean and empirical variance of the samples for the reconstructed signal $s$ (these are the target parameters of the problem). It is also common to show the empirical mean and empirical variance of the samples for latent fields that have an interpretable physical meaning, such as the density contrast $\delta$.

Finally, to connect with the generation of constrained samples in Wiener filtering (see Section 2.4), we note that each sample in the chain is one constrained realisation of the corresponding field, given the data (see Figure 4, second column). In contrast to Wiener filtering, where independent samples can be generated straightforwardly, samples along the chain are correlated for non-linear models.

# 5 Joint field-parameter sampling

This section extends the non-linear field-level inference framework for simultaneously inferring both the underlying physical fields and the cosmological parameters that govern them.

## 5.1 Posterior and its gradient for joint sampling

This section extends the non-linear field-level inference framework for simultaneously inferring both the underlying physical fields and the cosmological parameters that govern them.

We consider again the model described in Section 4.1, but leave the cosmological parameters $\{A_s, n_s, f_{NL}\}$ free. We aim to jointly infer the signal field and the cosmological parameters. The vector of target parameters is therefore $\theta \equiv \{A_s, n_s, f_{NL}, s\}$.

We now need to define the new (unnormalised) log-prior, log-likelihood, and log-posterior for the problem. For the prior, it is common to adopt a *separable* prior for the parameters and field, which means that

$$\ln p(\theta) = \ln p(A_s) + \ln p(n_s) + \ln p(f_{NL}) + \ln p(s). \tag{41}$$

For illustrative purposes, we choose Gaussian priors on each of the cosmological parameters. All assumptions made for the data-generating process remain the same as in Section 4.1. Therefore, the log-likelihood $\ln p(d|\theta)$ depends only on $\delta(\theta)$ and retains the form given by Equation (39), i.e.

$$\ln p(d|\theta) = -\frac{1}{2} [\delta(\theta) - d]^{\mathsf{T}} \mathbf{N}^{-1} [\delta(\theta) - d] + \text{const.} \tag{42}$$

Finally, the log-posterior is

$$\ln p(\theta|d) = \ln p(\theta) + \ln p(d|\theta) + \text{const.} \tag{43}$$

Depending on the sampling strategy (see Section 5.2), one may or may not need the gradient of the log-posterior with respect to the cosmological parameters, but the gradient with respect to the field is always needed. As before, gradients can be computed either analytically, through "manual" or automatic differentiation. Checking the gradient code against finite differences is always a useful sanity test.

## 5.2 Sampling strategy and implementation

In joint field–parameter inference problems, considerations regarding the optimal sampling strategy are important. Two main approaches can be distinguished, each with its pros and cons:

1. Sampling all parameters jointly in one block, in the same HMC/NUTS sampler. This is the strategy adopted in the Almanac code. Pros: no need to know conditionals; no need to design different samplers for different parameters. Cons: need for the gradient of the log-posterior with respect to all parameters (not only the field); difficulty in tuning the mass matrix when the signal field and other parameters have distinct natures (and therefore influence the data differently[9]), and which operate on disparate scales.

2. Using Gibbs sampling (see Section 3.2.1) to alternate between Gibbs updates of blocks of parameters (e.g. block-updating the field conditional on cosmological parameters, then cosmological parameters conditional on the field). More precisely, the sampling scheme will be Hamiltonian-within-Gibbs for the field, and "some-other-sampler-within-Gibbs" for cosmological/ancillary parameters. This is the strategy adopted in the BORG code. Pros: ease of implementation, since we usually know the conditional distributions; no systematic need for the gradient of the log-posterior with respect to other parameters (we can use, e.g., slice sampling [11] for other parameters, which is, in principle, rejection-free and does not require gradients). Cons: inability to take diagonal steps in the joint field–parameter space, which makes sampling inefficient when the field and parameters have a degenerate distribution.

Regardless of the sampling strategy, it is useful to parametrise the problem such that the target signal $s$ is a white-noise field and to "whiten" all other target parameters (i.e. rescale and shift them so that they typically take their values in $[-1, 1]$). This is essential if the first strategy is adopted, since all parameters need to be updated simultaneously by the proposal distribution.

## 5.3   Chain diagnostics and posterior summaries for fields and parameters

Generalising the considerations of Section 4.4, comprehensive MCMC diagnostics can be performed for the joint chains, including trace plots for individual parameters, the ESS, and the Gelman-Rubin $\hat{R}$ statistic. In addition to trace plots for selected field pixel values, one should check trace plots for each cosmological/ancillary parameter. It remains a good idea to start the chain in an overdispersed state (for the field) and check the sequential posterior power spectrum of the samples. The ESS gives the number of independent samples in the chain; it is especially important for posteriors of the cosmological parameters. We can also compute the PSRF $\hat{R}$ for the field and for the cosmological parameters. Caution is warranted: a chain can appear converged for some parameters even when it remains far from convergence for the field.

Communication of the posterior for the field takes the same form as before (e.g. visualisation of empirical means and variances for the signal and for latent fields). In addition, the posterior for cosmological/ancillary parameters (marginalised over values of the field) is usually presented as a corner plot (i.e., credible contours of the two-dimensional marginals for pairs of parameters; see Figure 5).

# 6   State-of-the-art applications

Without attempting to be exhaustive, this section highlights diverse applications of field-level inference across cosmological probes and showcases its impact on modern cosmological research.

---

[9]For example, in our problem: an infinitesimal change of the field value in one pixel of $s$ will only have a local effect on the data $d$, whereas an infinitesimal change of the primordial non-Gaussianity parameter $f_{NL}$ will have an effect everywhere.

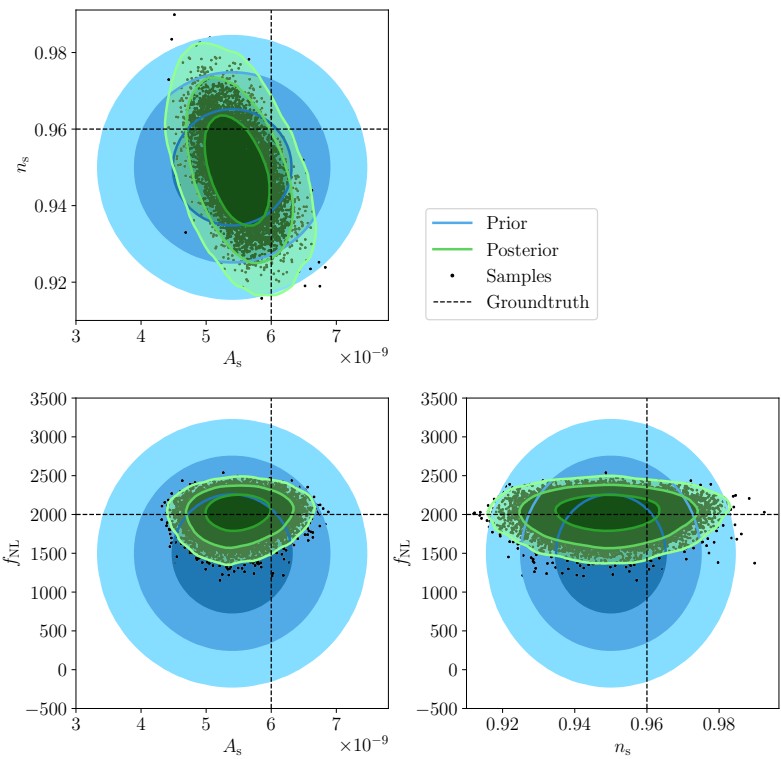

Figure 5: Triangle plot of cosmological parameters, marginalised over values of the field, obtained from sampling a joint field–parameter Bayesian hierarchical model.

## 6.1 Field-level inference for the cosmic microwave background

The cosmic microwave background (CMB) was the first field-level inference problem to be solved in cosmology [12], leveraging the assumption that the CMB map is a GRF, which allows for Wiener filtering. Under the assumption that the CMB is Gaussian, remaining questions include the efficient joint sampling of the field and its power spectrum (especially at large scales) via Gibbs sampling [e.g. 13, 14], and component separation [e.g. 15]. Non-linear data models can also tackle polarisation, CMB lensing [e.g. 16, 17], and the inference of primordial non-Gaussianity.

## 6.2 Field-level inference for galaxy clustering

Field-level inference is extensively applied to galaxy clustering to model the formation of the large-scale structure under gravity. The BORG (Bayesian Origin Reconstruction from Galaxies) algorithm [18, 19, 20, 21, 22], for instance, has been applied to reconstruct initial conditions and analyse large survey datasets, involving millions of parameters and requiring significant computational resources. Previous applications to real observational data include the SDSS-II main galaxy sample [19], the 2M++ galaxy catalogue [20, 21], and the SDSS-III BOSS survey [22]. The state-of-the-art Manticore-Local analysis [23] is the most advanced constrained realisation suite of the local Universe to date.

Various *gravity models* can be employed, including phenomenological density-field models such as Gaussian–linear (Wiener filter) and log-normal models, as well as structure formation models such as Lagrangian perturbation theory (LPT) and particle-mesh (PM) or COLA [24] models. For reasons explained in 4.2, an important requirement is the differentiability of these models. A crucial ingredient in field-level inference for galaxy clustering is the *galaxy*

*bias model*, which describes the relationship between dark matter density and galaxy number counts. Unlike traditional approaches such as correlation functions, field-level inference requires bias models that are voxel-based, sufficiently flexible, and without a scale cut. Galaxy bias for field-level inference is usually luminosity-dependent, non-local, and deterministic. Various functional forms, such as power-law and truncated power-law, have been used.

The field-level *likelihood function* specifies the probability distribution of observed galaxy counts given the signal and parameters, accounting for the stochasticity of the processes involved in galaxy formation. It can accommodate the observed super-Poissonian behaviour of the variance of galaxy counts in high-density regions. This function can be based on Gaussian, Poisson, negative-binomial, or generalised Poisson distributions. Another approach is Effective Field Theory for field-level inference, which uses a Gaussian likelihood across the entire Fourier grid, rather than in configuration space [25, 26].

Foregrounds and systematics can be handled by sampling the amplitude of known contaminations using templates, in a block-sampling scheme [27]. In addition, it is possible to report and marginalise over unknown foreground contaminations [22, 28]. Constrained initial conditions can be used to perform posterior resimulations, which provide, in particular, a view of the cosmic web [29, 30] and of our cosmic neighbourhood at small scales [31, 32, 33]. Beyond density reconstruction and inference of initial conditions, field-level inference for galaxy clustering enables cosmological tests, such as the Alcock-Paczyński expansion test [34], and allows one to constrain cosmological parameters, including primordial non-Gaussianity [35, 36].

## 6.3 Field-level inference for other/joint cosmological probes

The methodology of field-level inference has expanded its scope to a wide range of other cosmological probes. The treatment is as presented in Section 5: writing down a Bayesian hierarchical model for all probes considered, and jointly sampling the signal field and cosmological/ancillary parameters (either all together or in a block-sampling scheme).

Field-level inference with photometric galaxy clustering involves using individual galaxy redshift and density field sampling to infer the underlying signal from photometric observations, allowing for reconstructions even with large redshift uncertainties [37, 38]. Application of field-level inference to weak lensing can be performed either in tomographic bins, using two-dimensional maps, or in three dimensions. The `Almanac` algorithm [39, 40] performs joint map and power spectrum inference on the sphere, independent of the cosmological model, whereas the `BORGWL` algorithm [41, 42] performs three-dimensional map and cosmological parameter inference, while accounting for cosmic structure formation.

Previous applications of field-level analysis to the Lyman-$\alpha$ forest have utilised a data model for quasar spectra based on propagator perturbation theory (PPT) and the fluctuating Gunn-Peterson approximation (FGPA), or the emulation of hydrodynamical simulations [43, 44]. Inference of velocity fields can be performed using peculiar velocity tracers such as those in the CosmicFlows-1 to -4 data. The reconstruction is traditionally done via Wiener filtering [45, 46, 47, 48], but can also be achieved with Bayesian hierarchical models for a more comprehensive approach [49, 50, 51, 52].

## 6.4 Machine learning for field-level inference

The recent and ongoing integration of machine learning (ML) techniques is accelerating and enhancing the capabilities of field-level inference. As ML models are usually natively differentiable, they can be used within forward modelling. For instance, the problem of constructing a sufficiently accurate and flexible galaxy bias model has been addressed using a small physics-informed neural network [53, 54]. ML field-level emulators [55] can be used to construct accelerated forward models. These can be utilised to significantly accelerate the modelling of

complex phenomena, particularly at non-linear scales, the computational cost of which would otherwise be prohibitive.

ML can also be applied to solving the inverse problem of field-level inference. Novel machine learning approaches, such as score-based generative diffusion models [56], are being explored for approximate posterior sampling, offering potential computational advantages for exploring high-dimensional spaces. A neural optimiser can be integrated into field-level inference for tasks such as initial-condition searches [57]. Such an approach is capable of using non-differentiable simulators, thereby paving the way to non-linear field-level inference surpassing the requirement of a differentiable physics model.

# 7 Conclusion

The field of cosmology is entering an era of unprecedented data and complex models, making field-level inference a critical framework for connecting theory to observations. It reconstructs cosmological maps and infers parameters by using a full field-level likelihood, tackling very high-dimensional problems. While simple linear models utilise Wiener filtering, the complexity of non-linear field-level models necessitates advanced MCMC techniques, particularly gradient-based samplers, which efficiently explore high-dimensional posteriors and are significantly aided by automatic differentiation.

This methodology is currently applied to various cosmological probes, including the cosmic microwave background, galaxy clustering, weak lensing, and peculiar velocity tracers. A significant ongoing development is the integration of machine learning techniques. The synergy between Bayesian inference and machine learning is continually enhancing the efficiency and capabilities of field-level inference, cementing its role as a fundamental tool for extracting profound insights from ever-growing cosmological datasets.

# Acknowledgements

I thank the organisers of the CoBALt workshop and of the Les Houches summer school on the Dark Universe for the opportunity to prepare these lectures. This work was done within the Aquila Consortium. It benefited from feedback by Sébastien Renaux-Petel, Cyril Pitrou, and many doctoral researchers who attended the lectures.

**Funding information.** The author acknowledges financial support from the Agence Nationale de la Recherche (ANR) through grant INFOCW, under reference ANR-23-CE46-0006-01. This work was made possible by the programme CoBALt held at the Institut Pascal, Université Paris-Saclay, with the support of the programme "Investissements d'avenir" ANR-11-IDEX-0003-01, and by the École de physique des Houches summer school on the Dark Universe.

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
