# Peer review of "Field-level inference in cosmology"

_SciPost Physics Lecture Notes_

## Round 1 · Referee Report · Nicolas Cerardi (Referee 1) · 2025-10-3

Disclosure of Generative AI use

The referee discloses that the following generative AI tools have been used in the preparation of this report:

Generative AI was used on a few sentences for language edition.

Report

These lecture notes on field-level inference are very well prepared and written. The structure follows a logical path, presenting Bayesian inference methods for linear and non-linear models, and finally reviewing the state of the art in cosmological applications. The notes maintain a good balance between theoretical developments and practical considerations regarding implementation. I suggest a few minor revisions before publication.

Minor comments
Section 1.1. From my understanding, forward modelling can also be a source of bias (e.g. selection effects in building the mock catalogues, or loss of power at small scales due to CDM simulation methods). It is therefore not only backward modelling that may introduce bias. Is it considered subdominant with respect to the biases from backward modelling, or do you assume an ideal bias-free forward model (e.g. one that includes all relevant physics from structure formation to observation)?
Section 1.2. You state that “any method that involves compression of the map-level data is not included”. However, in Section 1.1 you defined field-level inference as the method that uses the “full mock and observed catalogues”. This is a bit unclear to me: a catalogue isn’t already a compressed representation of the observed field? Can you clarify whether catalogues are included in your definition?
Section 2.1, line 127. Could you explain why the equation to retrieve C is not recursive?
Figure 2 (and related text). Is there a way to compute burn-in length from the samples, or is it always a visual estimate? How is this handled when dealing with hundreds or more parameters? Also, it would be useful to indicate in the legend that different colors correspond to different chains.
Section 5.2. Maybe you could expand on why we usually know the conditionals for cosmological applications of Gibbs sampling?

Recommendation

Ask for minor revision

---

## Round 1 · Referee Report · Aritra Gon (Referee 2) · 2025-11-14

Disclosure of Generative AI use

The referee discloses that the following generative AI tools have been used in the preparation of this report:

I used Grammarly and Gemini (used on 14 November 2025) to improve sentence construction and weed out grammatical errors from my report.

Report

The lecture notes on field-level inference in cosmology are well-written and clear. The notes describe the differences between inferring parameters from summary statistics and field-level data, such as CMB maps or density fields. It then provides an overview of the Bayesian framework for parameter inference at the field level, starting from a linear model. It describes the basics of different Markov Chain Monte Carlo techniques. Lastly, it provides some examples for field-level inference applied to different cosmological observables.

The notes offer a pedagogical understanding of the subject, providing both the theoretical knowledge and showing applications in different cosmological scenarios. I only have a few minor suggestions that the author might include before publication.

Minor comments:

  1. It would be useful if the notes provided a bit more detail on how field-level inference is different from summary statistics. In particular, how that field-level inference jointly reconstructs the initial conditions as latent variables while simultaneously constraining cosmological parameters. This helps to understand why field-level inference aims to extract more information than summary-based approaches.

  2. A short discussion of the origin of the extra information in field-level inference would be beneficial. For example, whether the nonlinear evolution of the density field is better captured at the field level, or constraining the initial density field reduces marginalisation over late-time mode coupling, which improves parameter sensitivity.

  3. Readers may benefit from a note on the comparison between field-level inference and approaches based on higher-order statistics such as the bispectrum or trispectrum, or other kinds of statistics, like kNN or density-split.

  4. A short discussion on the practical and conceptual challenges of field-level inference would be useful. For example, the accuracy required of the forward model at the field level for capturing the nonlinear evolution across cosmological scales. The computational cost of full forward modelling, treatment of projection effects, and survey systematics at the field level.

Recommendation

Publish (meets expectations and criteria for this Journal)

---

## Editorial Decision

awaiting_resubmission